# Microfiltration Membranes Modified with Silver Oxide by Plasma Treatment

**DOI:** 10.3390/membranes10060133

**Published:** 2020-06-26

**Authors:** Joanna Kacprzyńska-Gołacka, Anna Kowalik-Klimczak, Ewa Woskowicz, Piotr Wieciński, Monika Łożyńska, Sylwia Sowa, Wioletta Barszcz, Bernadetta Kaźmierczak

**Affiliations:** 1Łukasiewicz Research Networks—Institute for Sustainable Technology, 6/10 Pułaskiego St., 26-600 Radom, Poland; anna.kowalik-klimczak@itee.radom.pl (A.K.-K.); ewa.woskowicz@itee.radom.pl (E.W.); monika.lozynska@itee.radom.pl (M.Ł.); sylwia.sowa@itee.radom.pl (S.S.); wioletta.barszcz@itee.radom.pl (W.B.); bernadetta.kazmierczak@itee.radom.pl (B.K.); 2Faculty of Materials, Science and Engineering, Warsaw University of Technology, 141 Woloska St., 02-507 Warsaw, Poland; piotr.wiecinski@gmail.com

**Keywords:** magnetron sputtering, AgO coatings, polyamide membranes

## Abstract

Microfiltration (MF) membranes have been widely used for the separation and concentration of various components in food processing, biotechnology and wastewater treatment. The deposition of components from the feed solution and accumulation of bacteria on the surface and in the membrane matrix greatly reduce the effectiveness of MF. This is due to a decrease in the separation efficiency of the membrane, which contributes to a significant increase in operating costs and the cost of exploitative parts. In recent years, significant interest has arisen in the field of membrane modifications to make their surfaces resistant to the deposition of components from the feed solution and the accumulation of bacteria. The aim of this work was to develop appropriate process parameters for the plasma surface deposition of silver oxide (AgO) on MF polyamide membranes, which enables the fabrication of filtration materials with high permeability and antibacterial properties.

## 1. Introduction

Polymer membranes have been widely used in wastewater treatment, the food industry, biotechnology and medicine [1]. The main limitation of polymer membrane exploitation in the filtration processes is biofouling consisting of the colonization of the surface and pores of membranes by microorganisms [2]. As a result of this phenomenon, the filtration process efficiency decreases over time, the transmembrane pressure increases, and the membrane selectivity changes [3]. The formation of biofilm on the surface of filtration membranes is mainly caused by bacteria that constitute the majority of microorganisms present in the filtered media, which enable the absorption of other organic particles forming extracellular polymeric secretions [4]. The resulting biofilm leads to the accumulation of inorganic particles and forms an irreversible layer of sediment on the membrane surface. The deposition of components from the feed solution and the accumulation of bacteria on the surface and inside the pores of the membrane significantly reduces the effectiveness of the microfiltration process. This is due to a decrease in membrane separation efficiency, which contributes to a significant increase in the operating costs of the filtration process. In recent years, there has been a great interest in the membrane modification processes to improve their resistance to the deposition of the components of the filtered medium and the accumulation of bacteria on their surface [5]. 

The most promising approach to reduce biofouling and inhibit the growth of bacteria is the modification of the membrane surface [6,7,8,9,10,11,12]. An effective method is the deposition of a coating on the surface of the membrane, which is a selective layer and allows the surface properties of the membrane to be shaped. One of the method is creating chitosan coating on the surface of polyamide membranes to improve separation and anti-fouling properties. It was found that such a modified surface-active layer of the polyamide membrane (PA) has excellent resistance to fouling [13]. Another interesting solution is direct deposition of PDMA-b-PMMA-b-PDMA copolymer micelles and gel formed from a mixture of polyvinyl alcohol and PDMA-b-PMMA-b-PDMA copolymer, in the form of an antifouling coating on the surface of polysulfone membranes, to improve the fouling resistance surface with protein solutions [14]. The literature presents a number of membrane modifications consisting in the formation of a coating on the surface in order to obtain resistance to organic and biological pollution [13,14,15,16,17,18,19,20]. Recently, new trends have arisen around the development and use of plasma surface engineering techniques for the modification and shaping functional properties of polymeric materials [21,22,23]. Magnetron sputtering (MS-PVD) technology is one of the plasma methods of surface engineering used to create a thin functional coating on the membrane surface to impart bactericidal properties. Polymeric membranes are fabricated from polymers characterized by low heat resistance, a bar of electrical conductivity and porosity. The deposition of metal coatings by magnetron sputtering on such materials requires maintaining a sufficiently low temperature and carrying out the process to avoid surface polarization. Such process conditions may have an adverse effect on maintaining the stability of the structure of the deposited material and on its adhesion to the substrate. It is also important to select the appropriate current-voltage parameters of the deposition process to prevent the damage of appropriate porosity of the membrane and the decrease in the filtration flux.

The main aim of the work was to select parameters of the MS-PVD process for the deposition of silver oxide (AgO) coating on the surface of polyamide microfiltration membrane in order to obtain antibacterial activity while maintaining the structure and filtration performance of the native membrane material. According to the authors, the application of magnetron sputtering technology is a pioneering solution enabling the production of functional thin films on the membrane surface. In the paper, the authors present the possibilities of using the magnetron sputtering technique to deposit silver oxide coating on the polymer microfiltration membrane surface with antibacterial and photocatalytic properties. Native and modified membranes were characterized in terms of structure and morphology. In this work, the bactericidal and photocatalytic properties of modified membranes was confirmed while maintaining their filtration properties. To the best of author knowledge, the modification of membranes by silver oxide coating is the first step in the design process of the magnetron sputtering modification technology of membrane filtration materials.

## 2. Materials and Methods 

### 2.1. Coatings Deposition

The AgO coatings were deposited by magnetron sputtering (MS-PVD) method using a Standard 3 device made by Łukasiewicz Research Networks—Institute for Sustainable Technology (Radom, Poland) equipped with a magnetron plasma source located on the side walls of the vacuum chamber. In the deposition process, magnetron plasma sources, including targets made of Ag (99.99% pure), were used. The diameter of the targets was 100 mm. The distance between the sample and the plasma source was 200 mm. The coatings were deposited at room temperature with a reactive gas atmosphere which consisted of a mixture of oxygen and argon (10% O_2_ 99.9999% pure and 90% Ar 99.9999% pure). The AgO coatings were deposited by changes to the source power in the range 80–500 W. The process time was 30 s. Technological processes were carried out without substrate polarization.

### 2.2. Structure Characterization

The structure characterization was performed using a Hitachi Su-8000 scanning electron microscope (SEM; Warsaw, Poland) equipped with an electron gun with cold field emission. This type of electron source provides very good resolution with a relatively low beam current, which is beneficial when observing materials sensitive to the electron beam, such as the membranes being analyzed. Observations were made using the secondary electron signal (SE). An additional conductive layer was not deposited on the sample.

### 2.3. Bactericidal Properties 

The antibacterial properties of surface-modified membranes were tested against Gram-negative bacteria (*Escherichia coli*) and Gram-positive bacteria (*Bacillus subtilis*). Prior to the microbiological tests, the membranes were sterilized with UV-C in laminar cabinet for 30 min. To prepare the inoculum, the subculture from the slant was suspended in 20 mL of sterile Luria broth (LB Agar Miller, VWR Chemical) culture medium, and then shaken at 2.21 Hz at 37 °C for 24 h. Next was the suspension of appropriately diluted bacteria in physiological saline buffer (0.3 M KH_2_PO_4_ cz.d.a, Chempur). After that, 10 mL of buffer slurry was filtered through the membranes and placed on Luria broth (LB) with agar plates and incubated at 37 °C for 24 h. All tests were repeated three times for each tested coating.

### 2.4. Photocatalytic Properties

The analysis of the photocatalytic properties of the membranes was based on the degree of methylene blue degradation (0.1% *v*/*v*) under the UV light. Polyamide membranes modified with one-component coatings were placed in petri dishes. A 20 cm^3^ volume of dye solution was transferred onto their surface Irradiation was carried out with the UV-A lamp, which was placed above the petri dishes. After 24, 48 and 72 h UV-irradiation, spectrophotometric measurements were made at the wavelength of 665 nm using a Hach DR 6000 spectrophotometer. The tests were repeated three times for each sample tested. The reference test was a dye after contact with the native membrane to take into account the potential effect of the dye on the membrane.

### 2.5. Filtration Properties

The permeate flux was determined by measuring the time required to filter deionized water (100 cm^3^) through the membrane (8 cm^2^) under defined transmembrane pressure (500 mbar). The deionized water characterized with the conductivity and pH of 5.3 μS/cm and 6.5, respectively. For this purpose, the laboratory “dead end” filtration set-up was used. The filtration properties of membranes were evaluated based on the permeate flux (Equation (1)):(1)$$J_{p}=\frac{V_{p}}{A\cdot t}$$
where, *J**_P_* is the permeate flux, dm^3^/(m^2^∙s); *V_P_* is the permeate volume, dm^3^; *A* is the membrane area, m^2^; and *t* is the time needed to receive a defined volume of permeate.

In addition, the stability of the AgO coating deposited on the membrane surface was tested using a vacuum filtration kit. The process was carried out at a pressure of 0.5 bar. Concentrations of silver (Ag) ions in such obtained filtrates were determined with an inductively coupled plasma mass spectrometer (ICP-MS; iCAP Q, Thermo Fisher Scientific, Radom, Poland). The detailed operating parameters for the ICP-MS measurements are summarized in Table 1. Prior to the determination of the Ag concentration, the samples were mineralized with 5 mL 65% HNO_3_ for 40 min at 160 °C using a DigiPrep Mini device (SCP Science, Radom, Poland).

## 3. Results and Discussion

### 3.1. Structure Characterization

The SEM images of the surface tested samples with AgO coating created with different power of magnetron source (P_M-Ag_ = 80, 270 and 500 W) are shown in Figure 1. In the case of the process which was carried out with the lowest magnetron power (P_M-Ag_ = 80 W), instead of a continuous coating on the membrane surface, spherical particles of AgO with diameter 25 nm without connection were found. Increasing the magnetron power to P_M-Ag_ = 270 W in the process of deposition increased the size of these particles to 200 nm—some of these began to combine with each other, which is visible in Figure 1c. Further increasing of the magnetron power to P_M-Ag_ = 500 W caused the particles to combine and create a continuous AgO coating on the membrane surface (Figure 1d). 

The conducted surface analysis showed that the deposited coatings are made of spherical nanoparticles of Ag, which increase their diameter by increasing the magnetron power and combine to form a continuous silver oxide coating. This phenomenon indicates the island nature of the AgO coating grown on a polymer substrate. This makes it possible to shape the structure of the obtained coating by changing the process parameters. Depending on the power of the magnetron, it is possible to place AgO material on the membrane surface in the form of nanoparticles not connected to each other or in the form of continuous and homogeneous coating. According to the authors, this is a very positive phenomenon that will enable the proper selection of the functional coating structure to the filtration membrane which does not disturb its separation properties.

### 3.2. Antibacterial Properties

The antibacterial activity of AgO-modified membranes was assessed against two representative bacteria: *Escherichia coli* (*E. coli*) and *Bacillus subtilis* (*B. subtilis*), respectively. The coatings deposited with each power of magnetron led to the complete inhibition of both *E. coli* and *B. subtilis* growth on the membranes which revealed no colonies on their surfaces after incubation on agar plates (Figure 2).

No significant effect of the AgO coating deposition parameters on the change in microbial survival was observed for both the Gram-negative and Gram-positive bacteria (Figure 3). The deposition of the coating with P_M-Ag_ = 80 W resulted in complete reduction of viability. The same was observed for the coatings deposited with higher power of magnetron.

These results may be associated with the structure of the coatings, which resulted from the MS-PVD technique. Silver oxide was formed as spherical nano-size particles on the surface of the membrane. In the literature, nanoparticles of AgO are considered as strong bactericidal agents similar to Ag nanoparticles [24,25,26]. Silver and silver oxide nanoparticles have the ability to anchor to and penetrate external bacterial structures causing the damage of respiration system and the permeability of the cell membrane. This consequently leads to the death of microorganisms [24,27]. The formation of free radicals was also proposed to explain cell death as a result of silver nanoparticles. Danilczuk et al. have found that silver nanoparticles generate free radicals in contact with bacteria, which was revealed using the electron spin resonance spectroscopy [26]. These radicals can damage the cell membrane leading to leakage of cytoplasm and cell death [28]. In turn, Woskowicz et al. showed that the strong bactericidal properties of the AgO coated by magnetron sputtering on polymer surface can be associated with low durability of the coating and the release of ions into the environment [22]. 

### 3.3. Photocatalytic Properties

The results of testing the photocatalytic properties of membrane covered with AgO coatings are shown in Figure 4. 

The analysis of organic dye showed that the best photocatalytic properties present the AgO coating deposited with magnetron power P_M-Ag_ = 80 W after 24 h. In this case the decreasing of absorbance around 30% was observed in comparison with control membrane. An almost total reduction of organic dye was found for the AgO coatings with magnetron power 80 W and 500 W after 72 h. In both cases, the decreasing of absorbance around 80% were noticed in comparison with native membrane. Results shows very good photocatalytic properties for these coatings. The smallest reduction of methylene blue was found for the AgO coating produced with magnetron power 270 W. The level of absorbance was comparable to the results obtained for the native membrane, which may indicate that this coating does not show photocatalytic properties. It can be concluded that the magnetron power has a significant impact on the efficiency of organic dye degradation. The high degree of dye reduction by the coating with the lowest applied magnetron power may be due to the fact that during the deposition process silver oxide is produced in the form of nanoparticles (Figure 1a). However, the photocatalytic properties of silver nanoparticles are already known in the literature [29,30,31,32] and depend on many factors (the method of nanoparticle synthesis, the type of dye and its concentration), Vanaja et al., in a study about silver nanoparticles, showed a reduction of methylene blue by approximately 30%, 60% and 90% after 24, 48 and 72 h of the process [33]. These results show the correlation with the results obtained in our research.

### 3.4. Filtration Properties and Stability of Coatings

The effect of plasma power during surface treatment of a polyamide membrane was examined on the permeate flux determined during filtration of demineralized water through the membranes. It was found that plasma surface treatment of the membrane using AgO carried out at the power of 80 W for 30 s did not change the permeate flux as it was compared to the permeate flux determined for native membrane (Figure 5). The surface of the membrane treated with plasma at 80 W was characterized by a polymer structure similar to that observed for the native membrane However, single AgO nanostructures were observed on the treated surface (Figure 1), which did not affect its filtration properties (Figure 5). Plasma treatment of the polyamide membrane carried out at 270 W and 500 W resulted in the decrease of the permeate flux by approximately 7% and 8%, respectively (Figure 5). This was caused by a significant thickening of the polymer structure resulting from the deposition of the AgO coating under the applied process conditions (Figure 1). In membrane filtration, it is preferable to keep high permeate flux enabling the desired efficiency of the wastewater treatment process; thus, it is preferable to use the lowest possible power of plasma treatment with the AgO of the polyamide membrane surface.

Based on the results of the structural, bactericidal, photocatalytic and filtration-separation tests obtained, a membrane with the magnetron power P_M-Ag_ = 80 W was selected for the stability tests. Stability was assessed based on the leaching test in H_2_O. The obtained results are presented in Figure 6.

The results of the test showed that the increased Ag concentration in the filtrate was observed only in the initial phase of test (for volume 100 mL) and it was in range 46.29 ± 0.91 µg/L. This can be due to the leaching of unbound silver particles from the surface of the deposited coating. In next step, the concentration of silver remained at a similar level and it was amounted average 0.72 ± 0.42 µg/L. It can be concluded that the membrane with an AgO coating is characterized by high stability of the coating under water conditions.

## 4. Conclusions

This research presents a novel interdisciplinary approach aimed at identifying innovative trends in the development of materials based on polymer membranes. The MS-PVD technique makes it possible to give new functional properties to a wide range of polymeric material, particularly in filtration membranes. In this study, the different magnetron powers of AgO were selected for polyamide membranes in order to obtain the structure, antibacterial properties, photocatalytic properties and filtration properties.

The surface analysis showed that the deposited coating is made of spherical particles of AgO, the diameters of which are increasing and combine in continuous form with increasing magnetron power. It is possible to shape the structure of the obtained coating by changing of the process parameters. The AgO coatings deposited at different magnetron powers caused the complete inhibition of grown colonies of *Escherichia coli* and *Bacillus subtilis*. It can be concluded that power also has a significant impact on the efficiency of organic dye decomposition. The best photocatalytic properties were found in the case of the AgO coating deposited with P_M-Ag_ =80 W. The photocatalytic properties did not exist in the case of the AgO coating with 500 W. The AgO coating deposited with 80 W did not change the permeate flux in comparison to the native membrane. A reduction by 7% and 8% of the permeate flux was observed for magnetron power 270 W and 500 W, respectively, compared to the native membrane. The membrane with AgO coating deposited with magnetron power 80 W was characterized by a high stability of the coating in the water conditions. An increased Ag concentration in the filtrate was observed only in the initial phase (for 100 mL of filtrated water). Further leaching steps indicated a similar level. The obtained results are promising and have potential in the application of surface-modified microfiltration membranes with AgO. Nevertheless, such produced materials should be tested under real operating conditions.

## Figures and Tables

**Figure 1 membranes-10-00133-f001:**
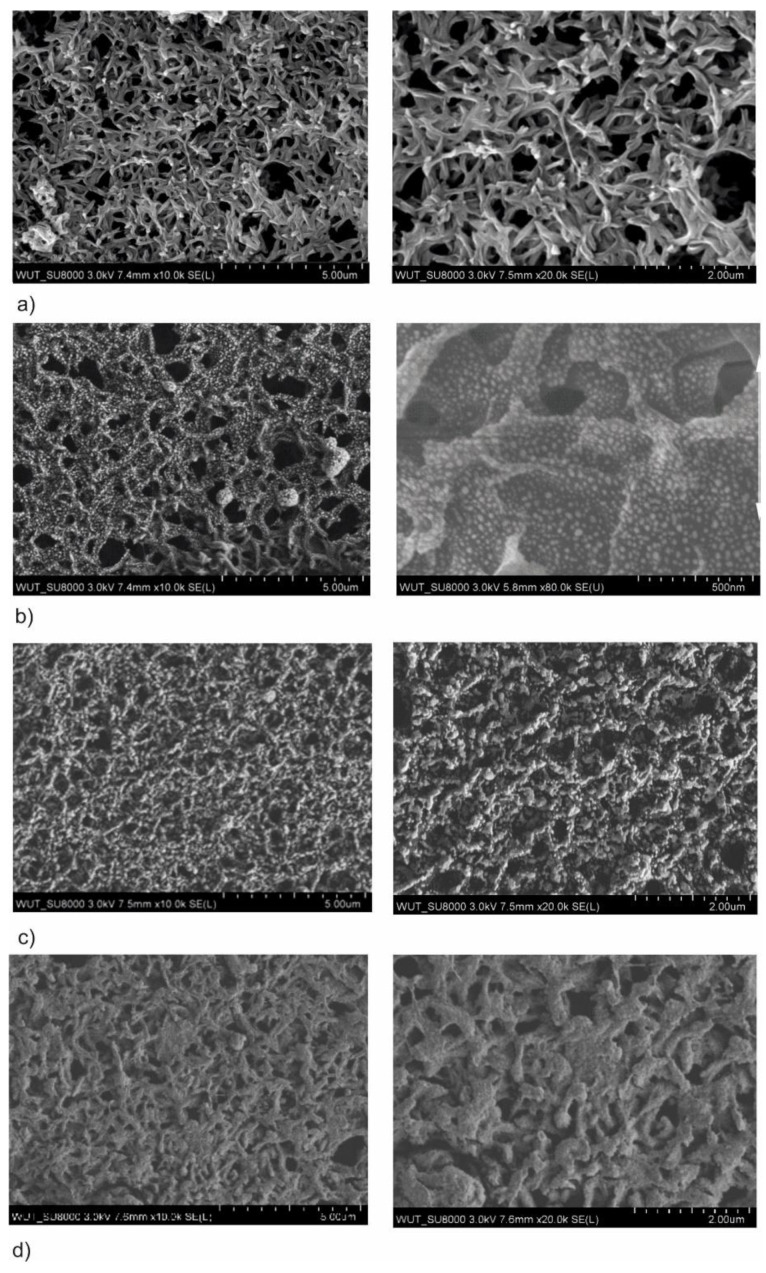
SEM images of membranes with the AgO coatings deposited at different magnetron powers P_M_: (**a**) native membrane; (**b**) P_M-Ag_ = 80 W, *t* = 30 s; (**c**) P_M-Ag_ = 270 W, *t* = 30 s; (**d**) P_M-Ag_ = 500 W, *t* = 30 s.

**Figure 2 membranes-10-00133-f002:**
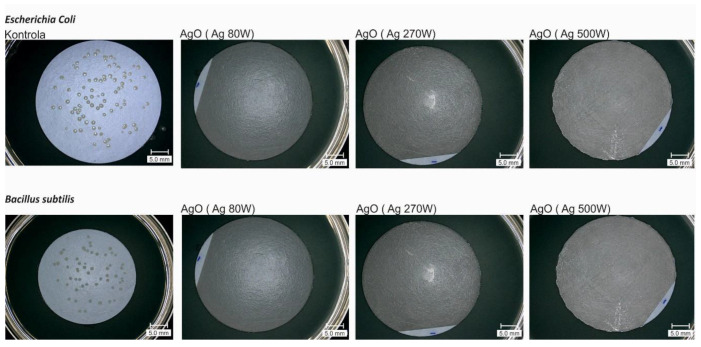
3D microscope images of the AgO coated membranes after filtration of bacterial suspensions.

**Figure 3 membranes-10-00133-f003:**
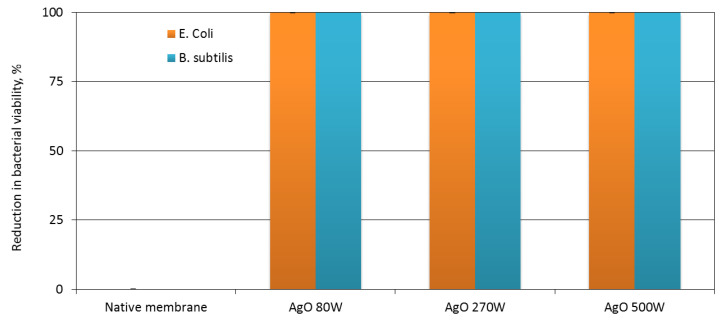
Reduction (%) in viability of *Escherichia coli* and *Bacillus subtilis* on the native membrane and membranes covered with an AgO coating deposited within 30 s at different magnetron powers P_M-Ag._

**Figure 4 membranes-10-00133-f004:**
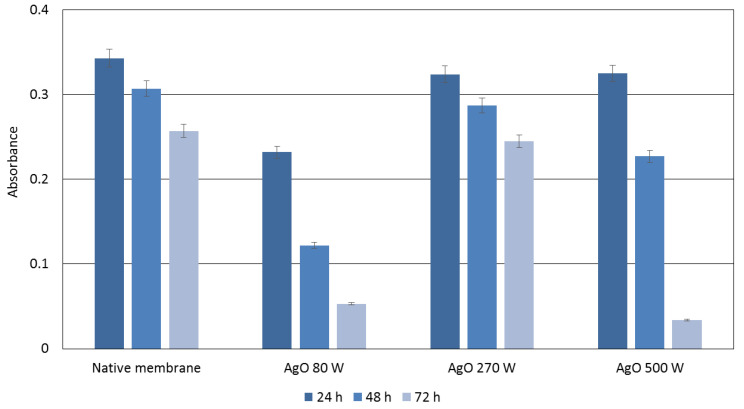
Comparison of the photocatalytic properties of native membrane and membranes covered with an AgO coating deposited within 30 s at different magnetron powers P_M-Ag_.

**Figure 5 membranes-10-00133-f005:**
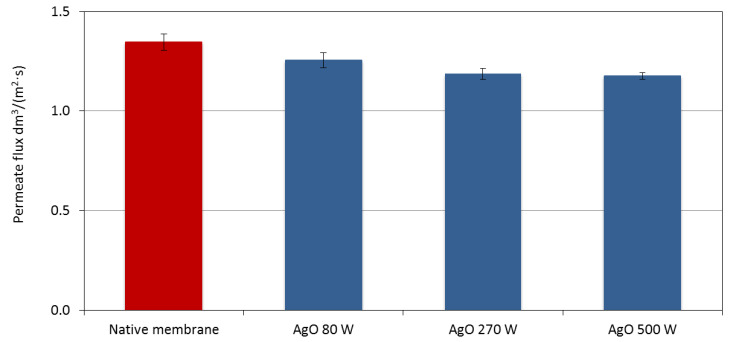
The effect of changing the magnetron power in the deposition process of the AgO coating on the membrane surface on the permeate flux determined during filtration of demineralized water.

**Figure 6 membranes-10-00133-f006:**
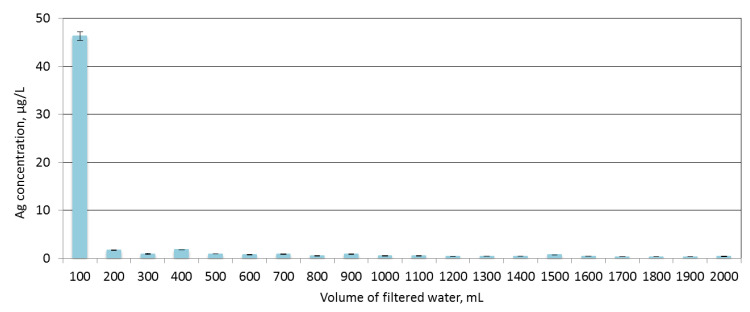
Concentrations of silver (Ag) ions in filtrate after the filtration process of demineralized water using membrane with the AgO coating (P_M-Ag_ = 80 W).

**Table 1 membranes-10-00133-t001:** Operating parameters of the inductively coupled plasma mass spectrometer (ICP-MS).

Parameters	Values
Forward power, W	1548.6
Cool gas flow, dm^3^/min	13.956
Auxiliary gas flow, dm^3^/min	0.8021
Nebulizer gas flow, dm^3^/min	1.02464
Dwell time, s	0.005
Number of replicates	3

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
