# Peer review of "Microfiltration Membranes Modified with Silver Oxide by Plasma Treatment"

_membranes, 2020, doi:10.3390/membranes10060133_

Round 1

Reviewer 1 Report

The manuscript by Kacprzyńska-Gołacka and co-workers describes membrane surface modification with AgO by plasma treatment. The research is timely and of interest to a broad audience working on water treatment and membranes. Therefore the topic fits well within the scope of the journal Membranes. However, there are some minor and major points that must be addressed prior to further consideration.

1) It is unnecessary to introduce abbreviations that are not used in the text. For instance EPS was introduced on page 1 and never used again.

2) The final paragraph of the introduction section should clearly state what the problems are with the current antifouling methods and how the proposed research solves the problem (hypothesis).

3) The first subsection of 2. Materials should be a section that lists all the chemicals, solvents, materials etc that were used during the study along with their supplier and grade/purity.

4) Report the stirring speed as well as the feed volume in the dead end cell. These are important experimental parameters that should be reported in the experimental section of the manuscript.

5) The authors should add some recent works on antibiofouling membrane development to cover diverse approaches (DOIs 10.1016/j.memsci.2019.117299, 10.1016/j.memsci.2020.118007, 10.1016/j.memsci.2020.118071).

6) The photocatalytic activity among other results reported, does not have error bars. Were the experiments repeated on independently prepared membranes? Details about reproducibility should be reported.

7) The scale bars for the smaller SEM images in Figure 1 are not legible. Enlarge them to make it clear for the readers.

8) The distance between the sample and the plasma source should be reported.

9) What is the actual amount of AgO in the final membrane (wt%)? This is essential information to include which might lead to a better understanding of the filtration performance.

10) How do the authors explain that the permeate flux is quasi the same for all membranes irrespective of coated or not?

11) Based on Figure 6, how does the Ag leaching from the membrane compare with the amount of Ag remaining in the membrane? In other words, what is the percentage of Ag the leached our during the 2000 mL wash?

Author Response

Thank you very much for the valuable suggestion and insightful comments contained in the review. Answers to the comments are presented below:

Ad.1: It is unnecessary to introduce abbreviations that are not used in the text. For instance EPS was introduced on page 1 and never used again.

Author response:

The authors agree with the reviewed suggestions that there is no need to write abbreviations that are not used in the text. The authors have removed the EPS abbreviation with.

Ad.2:The final paragraph of the introduction section should clearly state what the problems are with the current antifouling methods and how the proposed research solves the problem (hypothesis).

Author response:

As suggested by the reviewer, the authors corrected the introduction chapter.

Ad.3 The first subsection of 2. Materials should be a section that lists all the chemicals, solvents, materials etc that were used during the study along with their supplier and grade/purity.

Author response:

As suggested by the reviewer, the authors corrected the section 2 of the manuscript.

Ad.4 Report the stirring speed as well as the feed volume in the dead end cell. These are important experimental parameters that should be reported in the experimental section of the manuscript.

Author response:

As suggested by the reviewer, the authors corrected the experimental section of the manuscript.

Ad.5 The authors should add some recent works on antibiofouling membrane development to cover diverse approaches (DOIs 10.1016/j.memsci.2019.117299, 10.1016/j.memsci.2020.118007, 10.1016/j.memsci.2020.118071).

Author response:

The authors, as suggested by the reviewer, referred to the results of the research presented in the following publications: DOIs: 10.1016/j.memsci.2019.117299, 10.1016/j.memsci.2020.118007.

Ad.6 The photocatalytic activity among other results reported, does not have error bars. Were the experiments repeated on independently prepared membranes? Details about reproducibility should be reported.

Author response:

The lack of error bars in Figure 4 presenting the results of photocatalytic properties was associated with the error during conversion process. Tests of photocatalytic properties were repeated three times. The authors corrected Figure 4 and made corrections in the text as  suggested by the reviewer.

Ad.7 The scale bars for the smaller SEM images in Figure 1 are not legible. Enlarge them to make it clear for the readers.

Author response:

The authors corrected the scale bars in Figure 1 as suggested by the reviewer.

Ad.8 The distance between the sample and the plasma source should be reported.

Author response:

The authors supplemented the test with information about the distance between the sample and the plasma source

Ad.9 What is the actual amount of AgO in the final membrane (wt%)? This is essential information to include which might lead to a better understanding of the filtration performance.

Author response:

The quantitative assessment of AgO content in the membrane structure is very difficult. This is due to the fact that AgO is deposited in the form of a very thin coating only on the surface of the membrane and inside the pores in the surface layer of the membrane. According to the authors of quantitative AgO analyzes, it was possible to perform the basis of SEM analysis of morphology of cross-sections of modified membranes, which could provide a lot of information about the thickness of coatings deposited on the surface of membranes. However, these tests require the use of specialized research equipment and extensive analysis, which is the reason for carring out such tests in the future and present their results in another publication

Ad.10 How do the authors explain that the permeate flux is quasi the same for all membranes irrespective of coated or not?

Author response:

Figure 5 shows the results of the permeate flux has been corrected. Probably an error in Fig.5 was connected with file conversion process. The error bars shows the average values from 3 independently obtained membranes. The obtained permeate flux for unmodified and modified membranes is expected phenomenon. The parameters of the process were selected for modified coatings especially to have the least impact on the filtration properties of membranes in comparison for unmodified membranes. A small changes in the permeate flux are due to the island nature of the deposited coating. The authors’ experience shows that after using a higher magnetron power,  the homogenous coating will remain, which can lead to a significant reduction in the permeate flux.

Ad.11 Based on Figure 6, how does the Ag leaching from the membrane compare with the amount of Ag remaining in the membrane? In other words, what is the percentage of Ag the leached our during the 2000 mL wash?

Author response:

As the authors wrote in response 9, the quantitative assessment of AgO content in the membrane structure is very difficult. Similarly, it is difficult to assess the amount of AgO remaining on the membrane after the filtration process. Based on the conducted stability tests, it was estimated that during filtration 2000 ml of water was released jointly amount 60ug Ag per liter.

Reviewer 2 Report

In this work, the authors deposited silver oxide (AgO) on MF polyamide membranes by plasma surface deposition. The antibacterial property of the membranes were improved, however, there are still some experiments and comprehensive explanation should be provided. Thus, it should be carefully revised before this manuscript could be published on this journal.

Some comments are as follows:

  1. From the SEM images of membranes with AgO coatings, it is hard to know the AgO nanoparticles loading content in the membrane. some other characterization, e.g. XPS and TGA could be used to investigate the AgO content.
  2. SEM cross-sectional morphologies of the modified membranes should be provided, how about the AgO deposition uniformity and thickness on the membrane surface? This is important for improving the antifouling property.
  3. The authors have explained the membrane antibacterial property with silver (Ag) nanoparticles in manuscript line 166 and 199. However, in this paper, AgO nanoparticles were deposited on the membrane surface. The authors should carefully explain the antibacterial mechanism for the membranes.
  4. The antibacterial tests should be repeated at least three times, and the average value and error bar should be provided in Fig. 3, 4 and 6. Moreover, the plate count and inhibition zone could be provided to further investigate the membrane antibacterial performance.
  5. Fig. 5 just give out the membrane water permeation, how about the solute rejection after AgO nanoparticles depositing on the membrane surface? Please provide some data for it.

Author Response

Thank you very much for the valuable suggestion and insightful comments contained in the review. Answers to the comments are presented below:

Ad.1 and Ad.2

  1. From the SEM images of membranes with AgO coatings, it is hard to know the AgO nanoparticles loading content in the membrane. some other characterization, e.g. XPS and TGA could be used to investigate the AgO content.

2: SEM cross-sectional morphologies of the modified membranes should be provided, how about the AgO deposition uniformity and thickness on the membrane surface? This is important for improving the antifouling property.

Author response:

The authors agree with the reviewer's suggestion that the XPS and TGA technique could be used to investigate the AgO concentration in the membrane. The authors also agree that SEM cross-sectional morphologies of the modified membranes could provide a lot of information about the thickness of coatings deposited on the surface of membranes. However, these tests require the use of specialized research equipment and extensive analysis, which is the reason for carring out such tests in the future and present their results in another publication.

Ad.3 The authors have explained the membrane antibacterial property with silver (Ag) nanoparticles in manuscript line 166 and 199. However, in this paper, AgO nanoparticles were deposited on the membrane surface. The authors should carefully explain the antibacterial mechanism for the membranes.

Author response:

A silver oxide (AgO) coatings was used to modify the polymer membranes, because using of metallic silver for this purpose would cause a significant decrease of the permeate flux. One of the mechanisms of bactericidal action is the mechanism based on the release of ions. The mechanism of action on bacteria will be very similar for both cases when it was using silver nanoparticles and silver oxide nanoparticles [1,2]. However, the mechanism of the effect of silver oxide on bacteria was not the subject of research in our work, therefore in its description we use only available literature data.

[1] Prahbu S., Poulose E. K. Silver nanoparticles: mechanism of antimicrobial action, synthesis, medical applications, and toxicity effects. International Nano Letters 2012, 2, 1-10.

[2] Shen W., Li P., Feng H., Ge Y., Liu Z., Feng L. The bactericidal mechanism of action against Staphylococcus aureus for AgO nanoparticles. Material Science and Engineering C 2017, 75, 610-619.

Ad.4 The antibacterial tests should be repeated at least three times, and the average value and error bar should be provided in Fig. 3, 4 and 6. Moreover, the plate count and inhibition zone could be provided to further investigate the membrane antibacterial performance.

Author response:

Bactericidal tests were carried out on three independently modified polymer membranes for each tested coating.

In order to analyse bactericidal properties of membranes with AgO coatings, the authors developed their own research methodology, which was to closely reproduce the real conditions of using filter materials. This method involves filtering through modified polymer membranes an inoculum of tested bacteria of known concentration, suspended in physiological saline buffer. The authors supplemented chapter 2.3 with a more detailed description of the developed methodology.

Ad.5 Fig. 5 just give out the membrane water permeation, how about the solute rejection after AgO nanoparticles depositing on the membrane surface? Please provide some data for it.

Author response:

The authors as one of the ways to compare membrane properties before and after modification have chosen research permeate flux. Demineralized water was used for the research, while in the next step of research, which will be part of the next publication, it will be planned to examine the stream, use other medium and check both the permeate flux speed and the efficiency of treatment.

Round 2

Reviewer 1 Report

The manuscript has improved considerably. There are some minor editing issues left to do before publication:

1, in line 117 remove the space between the degree sign and C

2, line 119-120 has an editing issue with a backslash

3, the formatting of the unit in Figure 5 needs to be corrected

4, article numbers are missing in both references #19 and #20